# Cytomegalovirus Blood DNAemia in Patients with Severe SARS-CoV-2 Pneumonia

**DOI:** 10.3390/idr17010008

**Published:** 2025-01-26

**Authors:** Jean-Baptiste Mesland, Christine Collienne, Virginie Montiel, Alexis Werion, Philippe Hantson, Xavier Wittebole, Pierre-François Laterre, Ludovic Gerard

**Affiliations:** 1Critical Care Department, Cliniques Universitaires Saint-Luc, Université Catolique de Louvain, 1200 Brussels, Belgium; 2Department of Critical Care Medicine, CHR Mons-Hainaut, 7000 Mons, Belgium

**Keywords:** COVID-19, SARS-CoV-2, cytomegalovirus, CMV, mechanical ventilation

## Abstract

Introduction: Cytomegalovirus (CMV) DNAemia has been described in critically ill patients, including patients with severe acute respiratory syndrome-coronavirus2 (SARS-CoV-2) infection. Our objective is to evaluate the prevalence and clinical impact of CMV DNAemia among patients undergoing invasive mechanical ventilation (IMV) for severe SARS-CoV-2 infection and to explore the association between CMV DNAemia levels and clinical outcomes. Methods: In this retrospective monocentric study, we included patients admitted in a tertiary ICU for severe COVID-19 and who required IMV. We aimed to compare clinical and demographic variables between patients with and without CMV DNAemia. Univariate and Cox regression analyses were performed to identify factors associated with CMV DNAemia. Results: During the study period, CMV blood DNAemia occurred in 30/135 patients (22%). Patients with CMV blood DNAemia had longer ICU and hospital length of stay, as well as longer duration of IMV, and were more likely to have received dexamethasone. However, there was no significant difference in ICU mortality between patients with and without CMV DNAemia (64.8% vs. 56.7%, *p* = 0.42). The Cox regression analysis showed that dexamethasone was the only factor independently associated with CMV blood DNAemia (HR 4.23 [1.006–17.792], *p* = 0.049). Conclusions: In patients with severe SARS-CoV-2 pneumonia requiring IMV, CMV DNAemia is common and associated with prolonged ventilation and increased LOS but not with increased mortality.

## 1. Introduction

Cytomegalovirus (CMV) blood reactivation has previously been described in critically ill patients, including in non-immunocompromised populations [1]. It is still a matter of debate whether it is a hallmark of the immune status of the patients or if it plays a specific pathogenic role and therefore influences the clinical course and the outcomes [2]. Severe acute respiratory syndrome-coronavirus2 (SARS-CoV-2) infection has been associated with prolonged mechanical ventilation, which is a risk factor of CMV blood reactivation [3]. Therefore, we aimed to evaluate the prevalence and clinical impact of CMV DNAemia among patients undergoing IMV for severe SARS-CoV-2 infection and to explore the association between CMV DNAemia levels and clinical outcomes such as ICU and hospital length of stay.

## 2. Material and Methods

We performed a retrospective monocentric cohort study among patients with severe COVID-19 infection that required invasive mechanical ventilation (IMV) and who developed CMV DNAemia during their intensive care unit (ICU) stay. We screened all patients admitted from March 2020 to January 2022 with confirmed SARS-CoV-2 pneumonia based on reverse transcriptase polymerase chain reaction (RT-PCR) and who required IMV for acute respiratory failure. The clinical outcomes assessed included ICU mortality, ICU length of stay, hospital length of stay, duration of mechanical ventilation, and duration of ECMO support. Baseline characteristics such as Acute Physiology and Chronic Health Evaluation II (APACHE II) and Sequential Organ Failure Assessment (SOFA) scores were collected at ICU admission to evaluate patient severity. Patients were treated as recommended by international guidelines, which evolved over time. As systematic blood CMV PCR testing is not part of our clinical practice, we included all patients who had undergone at least one CMV PCR test during their ICU stay. Testing was performed at physician discretion based on clinical suspicion of CMV reactivation, which included prolonged ICU stay, worsening clinical status, or laboratory findings indicative of possible viral reactivation. Baseline CMV serostatus prior to SARS-CoV-2 infection was not available for this cohort. Detection was performed with quantitative PCR in whole blood by using an Alinity M CMV kit (Abbott, Chicago, IL, USA) and reported in IU/mL. The assay can quantitate CMV over the range of 30 IU/mL (1.48 Log IU/mL) to 100,000,000 IU/mL (8.00 Log IU/mL). To date, there is no recommendation of a specific threshold for diagnosing CMV reactivation in both blood and respiratory samples from patients in the ICU [4]. Virus load > 500 IU/mL, as used in previous studies [5,6], was used here for the definition of CMV DNAemia. This study was approved by the Ethics Committee of Cliniques Universitaires Saint-Luc, Brussel, Belgium (UCL N° 2021/15JAN/016).

Statistical analyses were performed using SPSS 21 software (SPSS software [IBM Corp. 2011. IBM SPSS Statistics for Windows, Version 21.0. Armonk, NY, USA: IBM Corp]). Values were expressed as median (+/− interquartile range) for continuous variables and counts (per percent of group) for qualitative variables. The data underwent Kolmogorov–Smirnov normality and Bartlett’s testing for homogeneity of variance. We compared clinical and demographic variables between patients having or not blood CMV reactivation using the Chi-squared test or Fisher’s exact as appropriate and the Mann–Whitney test for quantitative data, respectively. In order to find factors influencing the occurrence of blood CMV DNAemia (within 180 days of ICU admission), a Cox proportional hazards model was built as follows: all the variables significant in the univariable analysis were entered into a multivariable logistic regression with a backward elimination procedure based on the likelihood ratio. The results were expressed as hazard ratios (HRs) with 95% confidence intervals [IC95]. All tests were two-sided, with a significance level set at 0.05. The 180-day follow-up period was selected to capture both early and late CMV DNAemia events and assess their impact on long-term outcomes, as shorter follow-up periods may not fully reflect the prolonged disease course of severe COVID-19 patients.

## 3. Results

During the study period, 188 patients were admitted with severe SARS-CoV-2 pneumonia and required mechanical ventilation, of whom 135 underwent at least one CMV PCR test and were included in the study (Figure 1, Appendix A). Patient characteristics at baseline, in addition to adjuvant therapies, respiratory variables, and outcomes, are summarized in Table 1 and Table 2. CMV blood DNAemia occurred in 30/135 patients (22%), with a median highest value for CMV viral load of 4366 IU/mL [1250; 6775]. In the population of patients who had a viral load < 500 IU/mL, 52 patients had a truly negative viral load, 32 had a mildly positive viral load (<50 IU/mL) during their stay, and 21 had a maximum viral load between 50 and 500 IU/mL (187 IU/mL [88; 273]). Baseline characteristics were similar between groups. ICU and hospital mortality were similar between patients developing CMV DNAemia or not (17/30 vs. 68/105, *p* = 0.42 and 18/30 vs. 71/105, *p* = 0.44). Patients with CMV DNAemia had longer ICU length of stay (LOS) (68 days [47.5; 97.25] vs. 23 days [14; 47], *p* < 0.0001), hospital LOS (70 days [49.5; 114.5] vs. 31 days [20; 69.5], *p* < 0.0001), and duration of IMV (62.5 days [41; 82.8] vs. 20 days [12; 37], *p* < 0.0001). For patients treated with extra-corporeal membrane oxygenation (ECMO) therapy (*n* = 39), CMV DNAemia (*n* = 11) was associated with an increased ECMO duration (26 days [11; 47] vs. 54 days [34; 105], *p* = 0.0076).

Patients with CMV DNAemia were more likely to have received dexamethasone (28/30 vs. 77/105, *p* = 0.020). In addition, a Cox regression analysis (Table 3), adjusted for relevant covariables, revealed that dexamethasone was the only factor independently associated with CMV DNAemia in the first 180 days of ICU admission (adjusted hazard ratio [IC 95] 4.23 [1.006–17.792], *p* = 0.049).

The study of three distinct subgroups—the absence of CMV DNAemia, CMV DNAemia < 500 IU/mL, and CMV DNAemia > 500 IU/mL—yielded findings consistent with those previously reported (Appendix A).

CMV testing frequency was similar between groups (0.47 test/week [0.3; 0.7] vs. 0.44 test/week [0.29; 0.71], *p* = 0.67). During their ICU stay, patients had a median of two [1; 4] tests performed. Patients developing CMV DNAemia had their first positive PCR at a median of 18 days [11.75; 25.75] from ICU admission and their first PCR > 500 IU/mL at a median of 36 days [25.75; 50.75]. Valganciclovir was initiated in 16/30 patients (Appendix A) at the discretion of the treating physician. Patients who received valganciclovir treatment were younger (57 years [51; 65] vs. 66 years [62; 75], *p* = 0.0061), had higher CMV DNAemia levels (6293 IU/mL [2831; 8029] vs. 1759 IU/mL [839; 4419], *p* = 0.0060), and included a higher proportion of patients on ECMO (10/16 vs. 1/14; *p* = 0.0017). Treatment was not associated with ICU mortality (9/16 vs. 8/14, *p* = 0.96).

## 4. Discussion

In our analysis, we found that 22% of patients undergoing IMV for severe SARS-CoV-2 infection developed significant CMV DNAemia (>500 IU/mL). To the best of our knowledge, this is the first study that focused on CMV DNAemia in patients under invasive MV for SARS-CoV-2 pneumonia. Recently, using the same thresholds for CMV reactivation (Appendix A), Gatto et al. [7] reported that 11.6% of patients admitted to the ICU for severe SARS-CoV-2 (mechanically ventilated or not) developed CMV blood reactivation. In addition, in a previous study involving patients under MV for ARDS unrelated to COVID-19, CMV reactivation was found in 18.4% (74/401) of patients.

In our cohort, CMV DNAemia was more frequent in patients who had received dexamethasone and was associated with prolonged MV duration and ICU and hospital LOS but not with mortality. The association between corticosteroids and CMV blood reactivation is still a matter of debate, with conflicting results in the literature [7,8]. Recently, Tassaneeyasin et al. described a cohort of 185 patients admitted to intensive care, whether ventilated or not. They reported the same association between CMV DNAemia in COVID-19 patients and the cumulative doses of dexamethasone received [9]. Conversely, Boers et al. described a cohort of 156 patients who underwent bronchoalveolar lavage as part of their management for COVID-19 to investigate viral or bacterial co-infections. Only 9 out of 156 patients had a viral load >10,000 copies/mL (in the BAL fluid), with no significant difference in the use of corticosteroids between those with or without significant reactivation [10]. Other immunomodulating therapies like tocizilumab were not used in our center, and therefore we were unable to study the effect of their use on the occurrence of CMV DNAemia.

Several studies have shown an association between CMV viral reactivation and duration of ventilation [1], ECMO [11], or ICU LOS [12], thus supporting our results. However, the association of CMV reactivation with mortality is still debated in the SARS-CoV-2 population [7,12,13], as well as in the general ICU population [6].

CMV reactivation encompasses not only viremia but also the potential development of tissue-invasive diseases, such as pneumonitis, colitis, esophagitis, encephalitis, and retinitis [14]. However, in critically ill patients with COVID-19 pneumonia, diagnosing CMV tissue-invasive disease with biopsies is complex, with the potential risk outweighing the benefit. In this context, treatment is often initiated without evidence of invasive pathology, solely based on high DNAemia. In our cohort, with a small number of treated patients, no difference in mortality was observed between patients treated or not. In a randomized, placebo-controlled trial examining pre-emptive therapy in ventilated ICU patients with evidence of CMV reactivation, Papazian et al. randomized 76 patients to ganciclovir versus placebo. The trial was stopped early because of no significant difference in ventilator-free or mortality days within 60 days after randomization [15].

The absence of a difference in mortality between patients with or without CMV DNAemia, or between those with or without treated suspected reactivation, raises the question of whether CMV DNAemia is pathogenic in COVID-19. The pathogenicity of CMV reactivation in COVID-19 disease is still a matter of debate. Some experts believe that CMV reactivation is simply a marker of the severity of underlying disease, while others believe that it can be a direct cause of organ damage [6]. Indeed, CMV DNAemia could indirectly contribute to morbidity and mortality by modulating immune function and the inflammatory response [14]. This could lead to bacterial superinfection or the aggravation of inflammatory processes such as ARDS [16,17]. Further research is needed to answer this question and determine whether specific antiviral therapies can decrease morbidity and mortality in patients with CMV reactivation.

CMV is part of the Herpesviridae family, which comprises eight viruses potentially pathogenic to humans: herpes simplex virus type 1 and type 2 (HSV-1 and HSV-2), the varicella-zoster virus (VZV), cytomegalovirus (CMV), the Epstein–Barr virus (EBV), and the sixth, seventh, and eighth human herpesviruses (HHV-6, HHV-7, and HHV-8) [18]. Following primary infection, all these viruses can reactivate under specific acute conditions, most often facilitated by immunosuppression. Although all these viruses have been studied in the context of COVID-19, most of the research has focused on HSV and the EBV in critically ill patients. Based on bronchoalveolar lavage samples, Boers et al. reported an HSV reactivation incidence of 37% [10], while Giacobbe et al. found HSV reactivation in 29% of their patients using a similar threshold (viral load > 10⁴ copies/mL) [19]. They also demonstrated an association between viral loads in BAL samples and increased mortality. Regarding EBV reactivation, Guilouiller et al. reported a 54.3% incidence in blood samples (PCR) [20]. Mortality at 28 and 90 days was not significantly different between patients with and without reactivation. However, the duration of mechanical ventilation, as well as the incidence of ARDS, infections, and septic shock, was higher in the group of patients with EBV reactivation. Although the underlying mechanisms of these reactivations remain poorly understood, these findings highlight the importance of monitoring Herpesviridae reactivations beyond CMV and assessing the potential benefits of targeted antiviral therapy in this high-risk population.

Our study has several limitations that should be acknowledged. First, it is a retrospective and monocentric study with a limited number of patients, which may limit the validity of our analysis. Further studies should try to include a higher number of patients prospectively and in multiple centers. Second, CMV blood screening was not conducted in a protocolized manner and was left at the appreciation of the treating physician. Therefore, it could constitute a selection bias, as the sickest patients could have been more likely tested for CMV reactivation. To mitigate selection bias, future studies should implement systematic and standardized CMV screening protocols, ensuring that all eligible patients are tested, regardless of perceived clinical severity. Third, without knowledge of the baseline CMV serologic status, it becomes challenging to differentiate true reactivation from an acute infection (which is unlikely in our situation). We used the term CMV DNAemia rather than reactivation throughout our manuscript and in the description of our findings for this purpose. Fourth, there is a potential bias introduced by longer ICU stays in CMV-positive patients. Despite an equal frequency of CMV testing between groups, longer ICU stays increase the likelihood of reactivation due to sustained critical illness and immunosuppression. This confounding factor may partly explain the association between CMV DNAemia and longer ICU stays observed in our cohort. Finally, as we lack information on the SARS-CoV-2 variant infecting each patient, we were unable to assess the potential association between the infecting variant and CMV DNAemia.

## 5. Conclusions

CMV DNAemia (>500 IU/mL) is associated in our cohort with prolonged ventilation duration and increased LOS but not with increased mortality. These findings underscore the need for further studies to clarify the role of CMV reactivation in critically ill COVID-19 and non-COVID-19 patients.

## Figures and Tables

**Figure 1 idr-17-00008-f001:**
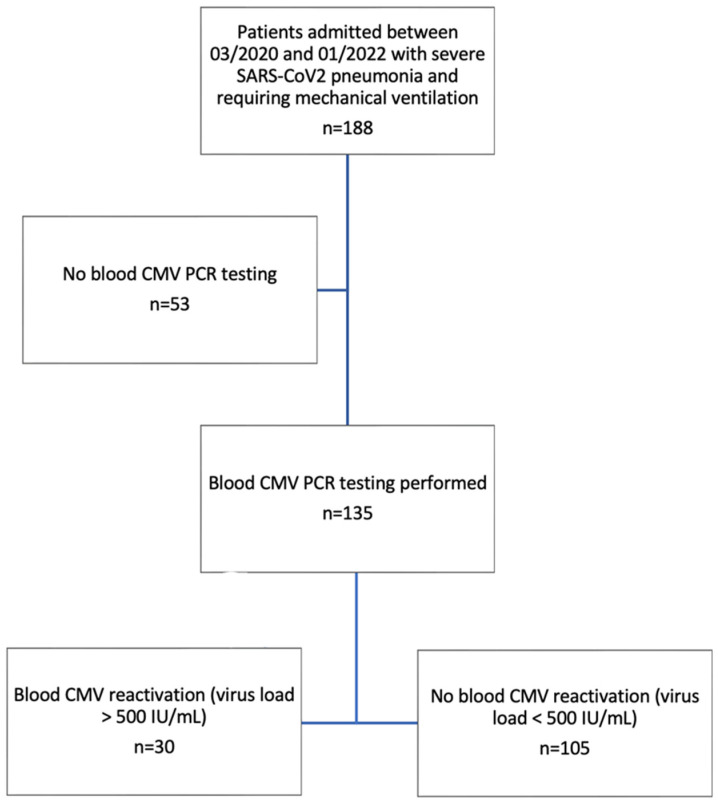
Study cohort flow chart.

**Table 1 idr-17-00008-t001:** Characteristics between patients developing or not cytomegalovirus DNAemia.

	CMV Negative (*n* = 105)	CMV Positive (*n* = 30)	*p* Value
Baseline Characteristics			
Age, years	63 (54; 70)	64 (55; 69)	0.97
Male (%)	75/105 (28.6)	10/30 (33.3)	0.61
BMI	28 (25; 32)	30 (26; 32)	0.57
Hypertension (%)	54 (51.4)	21 (70)	0.071
Diabetes mellitus (%)	37 (35.2)	8 (26.7)	0.38
Cardiomyopathy (%)	22 (21)	4 (13.3)	0.35
Chronic kidney disease (%)	10 (9.5)	2 (6.7)	0.63
Immunosuppression (%)	20 (19.1)	6 (20)	0.91
COPD (%)	7 (6.7)	0 (0)	0.15
Neoplasia <2 years (%)	9 (8.6)	0 (0)	0.097
SOFA	6 (4; 7)	6 (5; 8)	0.55
APACHE II	15 (13; 18)	14 (12; 19)	0.28
Biological parameters			
Admission CRP, mg/dL	139 (90; 216)	145 (65; 212)	0.66
Highest CRP, mg/dL	331 (266; 380)	344 (303; 413)	0.21
Admission LDH, IU/L	546 (440; 657)	485 (398; 563)	0.10
Highest LDH, IU/L	699 (583; 934)	712 (533; 1194)	0.85
Admission lymphocytes,/mcL	550 (385; 940)	610 (368; 920)	0.75
Lowest lymphocytes,/mcL	370 (190; 540)	290 (165; 420)	0.14
Admission platelet, 10^3^/mcL	222 (169; 306)	232 (157; 277)	0.89
Highest D-dimer, ng/mL	1554 (915; 5840)	1253 (580; 2562)	0.025
Admission fibrinogen, mg/dL	636 (503; 802)	670 (526; 793)	0.59
PaO_2_/FiO_2_ (admission)	68 (53; 84)	64 (54; 78)	0.59
Lowest PaO_2_/FiO_2_	48 (42; 51)	49 (45; 53)	0.23
Treatment			
Dexamethasone (%)	77 (73.3)	28 (93.3)	0.020
Hydroxychloroquine (%)	23 (21.9)	3 (10)	0.15
ECMO (%)	29 (27.6)	11 (36.7)	0.34
RRT (%)	28 (26.7)	13 (43.3)	0.08

CMV: cytomegalovirus, BMI: body mass index, COPD: chronic obstructive pulmonary disease, SOFA: sequential organ failure assessment, APACHE II: Acute Physiology and Chronic Health Evaluation II, ECMO: extra-corporeal membrane oxygenation, RRT: renal replacement therapy.

**Table 2 idr-17-00008-t002:** Patient outcomes between patients developing or not cytomegalovirus DNAemia.

	CMV Negative (*n* = 105)	CMV Positive (*n* = 30)	*p* Value
ICU mortality (%)	71 (64.8)	17 (56.7)	0.42
ICU length of stay, days	23 (14; 47)	68 (48; 98)	<0.0001
Hospital length of stay, days	31 (20; 70)	70 (50; 115)	<0.0001
Duration of ventilation, days	20 (12; 37)	62.5 (41; 83)	<0.0001
Duration of ECMO, days (*n* = 40)	26 (11; 47) (*n* = 29)	54 (34; 105) (*n* = 11)	0.0076

CMV: cytomegalovirus, ICU: intensive care unit, ECMO: extra-corporeal membrane oxygenation.

**Table 3 idr-17-00008-t003:** Cox regression analysis for variables associated with CMV DNAemia (within 180 days of ICU admission) in 135 patients receiving invasive mechanical ventilation for severe COVID-19.

Variables	Univariate			Multivariate		
	Hazard Ratio	IC 95	*p*	Adjusted Hazard Ratio	IC 95	*p*
SOFA score	0.91	0.736–1.125	0.38	0.915	0.744–1.126	0.403
Diabetes	0.509	0.206–1.262	0.145	0.593	0.261–1.343	0.21
Immunosuppressive therapy	1.188	0.438–3.219	0.735	1.164	0.46–2.941	0.75
Chronic kidney disease	0.877	0.181–4.261	0.87	0.875	0.18–4.247	0.87
Dexamethasone	6.047	1.31–27.87	0.021	4.23	1.006–17.792	0.049
Male sex	0.642	0.271–1.524	0.31	0.645	0.285–1.46	0.392
Age	1.022	0.988–1.058	0.21	1.014	0.982–1.046	0.394
Body mass index	0.998	0.94–1.059	0.94	0.998	0.94–1.059	0.937

SOFA: Sequential Organ Failure Assessment.

## Data Availability

The data presented in this study are available on request from the corresponding author.

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
