# Peer review of "Cytomegalovirus Blood DNAemia in Patients with Severe SARS-CoV-2 Pneumonia"

_2036-7449, 2025, doi:10.3390/idr17010008_

Round 1
Reviewer 1 Report
Comments and Suggestions for Authors
Cytomegalovirus (CMV) is a ubiquitous human pathogen because it establishes latent infection, lasting for the host's lifetime. Occasional CMV reactivation requires constant immunosurveillance by a fully functional immune system. It is still unclear how other infections, such as those with SARS-CoV-2, affect the immune control of CMV. In this manuscript, Mesland et al. investigated whether severe SARS-CoV-2 infection restrains immune control of CMV, thus allowing it to reactivate and induce co-morbidity. They found increased levels of CMV DNA in the blood of 30/135 patients with severe COVID-19 admitted to a tertiary ICU. Increased CMV blood DNAemia was associated with the dexamethasone treatment, and those patients had longer ICU and hospital lengths of stay.
This timely and relevant study reports clinically relevant findings. However, the current manuscript contains several issues that need to be addressed.
Major comments
1. The authors are aware of the issue in the study design, which is the unknown baseline CMV status. However, they should not hide this issue until the end of the discussion (lines 163-167) but mention it already in the material and methods while describing their cohort and try to incorporate it in the study design.
2. Along those lines, due to the unknown CMV status of the recruited patients, the authors cannot address one of their aims: “to evaluate the incidence of CMV DNAemia among patients undergoing IMV for severe SARS-CoV-2 infection”. Hence, they should rephrase their aim to deliverable study aspects.
3. One way to solve the above-mentioned issues is to use CMV DNA PCR to stratify the patients into different groups. In the current version of the manuscript, the authors compared two groups of patients, the “CMV negative” and “CMV positive” groups. However, 50% of patients in “CMV negative” groups were also CMV positive, albeit at lower levels (<500 IU/mL) than in the current “CMV positive” patients (>500 IU/mL) (lines 72-76). The study would benefit greatly if the authors would compare 3 groups: (i) COVID patients without CMV DNAemia, (ii) COVID patients with detected CMV DNA but without evidence of CMV reactivation (<500 IU/mL), and (iii) COVID patients with detected CMV DNA but with evidence of CMV reactivation (>500 IU/mL). In this way, the authors could better address important questions, such as whether there were differences in duration or dosing of dexamethasone treatment between patients with low- and high-CMV DNAemia. The other possibility could be to make correlations between investigated parameters and the detected CMV levels.
4. It would be important also to describe if there are any differences between CMV-positive patients (>500 IU/mL) who received valganciclovir to those who did not receive this treatment.
Minor comments
1. Table 2 suggests that all patients were treated with ECMO, which is not the case as the text suggests otherwise (lines 82-84). Please include the data on how many patients in each group were treated with ECMO.
2. Please add information, if possible, on the SARS-CoV-2 variants the patients were infected with.
Author Response
Major comments
- The authors are aware of the issue in the study design, which is the unknown baseline CMV status. However, they should not hide this issue until the end of the discussion (lines 163-167) but mention it already in the material and methods while describing their cohort and try to incorporate it in the study design.
Response : Thank you for your suggestion. We have added in Materiel and methods: “Testing was performed at physician discretion. Baseline CMV serostatus prior to SARS-CoV-2 infection was not available for this cohort. »
- Along those lines, due to the unknown CMV status of the recruited patients, the authors cannot address one of their aims: “to evaluate the incidence of CMV DNAemia among patients undergoing IMV for severe SARS-CoV-2 infection”. Hence, they should rephrase their aim to deliverable study aspects.
Response :
Thank you for your comment. We have rephrased our aim in this manner : “we aimed to evaluate the prevalence and clinical impact of CMV DNAemia among patients undergoing IMV for severe SARS-CoV-2 infection, and to explore the association between CMV DNAemia levels and clinical outcomes such as ICU and hospital length of stay.”
- One way to solve the above-mentioned issues is to use CMV DNA PCR to stratify the patients into different groups. In the current version of the manuscript, the authors compared two groups of patients, the “CMV negative” and “CMV positive” groups. However, 50% of patients in “CMV negative” groups were also CMV positive, albeit at lower levels (<500 IU/mL) than in the current “CMV positive” patients (>500 IU/mL) (lines 72-76). The study would benefit greatly if the authors would compare 3 groups: (i) COVID patients without CMV DNAemia, (ii) COVID patients with detected CMV DNA but without evidence of CMV reactivation (<500 IU/mL), and (iii) COVID patients with detected CMV DNA but with evidence of CMV reactivation (>500 IU/mL). In this way, the authors could better address important questions, such as whether there were differences in duration or dosing of dexamethasone treatment between patients with low- and high-CMV DNAemia. The other possibility could be to make correlations between investigated parameters and the detected CMV levels.
Response : Thank you for your comment.
Indeed, we had conducted these analyses without including them in the article to avoid overloading the main message. We have now added the comparison of the three groups in the Supplementary Material: Table 2. Despite a seemingly well-balanced population across the three groups, there remains a difference in the use of dexamethasone between the group with CMV DNAemia > 500 IU/mL and the group with no CMV DNAemia (28/30 vs. 37/52, p = 0.017), as well as between the group with CMV DNAemia > 500 IU/mL and the group with CMV DNAemia < 500 IU/mL (28/30 vs. 40/53, p = 0.042). Moreover, other main messages of the article is reflected in the subanalysis of these three subgroups: there is no significant difference in mortality between the three subgroups. However, there is an association between the development of CMV DNAemia and the duration of ventilation and hospitalisation, which is particularly marked in the group with CMV DNAemia > 500 IU/mL.
We have added the following statement in the results section:
"The study of three distinct subgroups—absence of CMV DNAemia, CMV DNAemia < 500 IU/mL, and CMV DNAemia > 500 IU/mL—yields findings consistent with those previously reported (Supplementary Material, Table 2)."
- It would be important also to describe if there are any differences between CMV-positive patients (>500 IU/mL) who received valganciclovir to those who did not receive this treatment.
Response :
We fully agree that including these data is of interest.
Once again, to avoid overloading the article, we have added this information to the Supplementary Material, Table 3. Given the small number of patients, the conclusions drawn from this analysis are limited and primarily descriptive. Patients who received valganciclovir treatment were younger (57 years [51; 65] vs. 66 years [62; 75], p = 0.0061), had higher CMV DNAemia levels (6293 IU/mL [2831; 8029] vs. 1759 IU/mL [839; 4419], p = 0.0060), and included a higher proportion of patients on ECMO (10/16 vs. 1/14; p = 0.0017).
We have added the following statement to the Results section:
"Patients who received valganciclovir treatment were younger (57 years [51; 65] vs. 66 years [62; 75], p = 0.0061), had higher CMV DNAemia levels (6293 IU/mL [2831; 8029] vs. 1759 IU/mL [839; 4419], p = 0.0060), and included a higher proportion of patients on ECMO (10/16 vs. 1/14; p = 0.0017)."
Minor comments
- Table 2 suggests that all patients were treated with ECMO, which is not the case as the text suggests otherwise (lines 82-84). Please include the data on how many patients in each group were treated with ECMO.
Response : Thank you for this comment. The information was already in table 1 but it has now been added in table 2 with (n=…) to make the information clearer.
- Please add information, if possible, on the SARS-CoV-2 variants the patients were infected with.
Response : We thank you for this again insightful suggestion.
Unfortunately, we do not have information on the infecting variant for each patient, particularly because, during the initial waves of COVID, identifying the specific variant was not part of our laboratory’s routine practice. While we could have inferred the infecting variant based on the admission date, assuming the patient was infected by the most prevalent variant circulating in our country at that specific time, we felt that relying on potentially incorrect assumptions would not be appropriate.
We have therefore added the following statement to the limitations section:
"Finally, as we lack information on the SARS-CoV-2 variant infecting each patient, we were unable to assess the potential association between the infecting variant and CMV DNAemia."
Reviewer 2 Report
Comments and Suggestions for Authors
In this retrospective monocentric study, the authors describe the outcome of ICU patients (admitted due to severe COVID-19) with and without CMV DNAemia. Although this is not the first study investigating this, I found it interesting to read. The data are presented very concise and well and I think they are of interest for the journals´ readers or physicians working in the field. The authors confirm an absence of a difference in additional morbidity and mortality between ICU patients with or without CMV DNAemia.
I have a few points, that might be addressed:
1) The unpaired t-test was used to calculate quantitative data. If dealing with the viral loads or other not evenly distributed data, I would rather recommend a test for non-parametric data, (e.g. Mann-Whitney test).
2) Patients with CMV DNAemia had longer ICU length of stay (LOS) (68 days vs 23 days, p < 0.0001). Do the statistics account for the bias, that the chance to discover CMV-DNAemia is higher due to more testing or the fact that biologically it is more likely that a reactivation takes place the longer the patient is critically ill? This should at least be discussed as another important bias/ limitation.
3) If feasible a bit more information (or visualization) on the viral load levels (peak, duration of DNAeamia) over time would be very interesting, to evaluate how significant the reactivations were. Is it possible to stratify the patienst not only according to negative vs. positive DNAemia but also in high and low CMV-DNAemia?
4) I understood that there are only little data on BAL-fluid PCR results and these were thus not added to statistics?
Author Response
I have a few points, that might be addressed:
- The unpaired t-test was used to calculate quantitative data. If dealing with the viral loads or other not evenly distributed data, I would rather recommend a test for non-parametric data, (e.g. Mann-Whitney test).
Response : We thank you for your comment and fully agree with your observation.
Upon review, we realized that we had actually performed a Mann-Whitney test for all these data but had mistakenly mentioned the unpaired t-test in our manuscript.
We have now corrected this in the Materials and Methods section.
- Patients with CMV DNAemia had longer ICU length of stay (LOS) (68 days vs 23 days, p < 0.0001). Do the statistics account for the bias, that the chance to discover CMV-DNAemia is higher due to more testing or the fact that biologically it is more likely that a reactivation takes place the longer the patient is critically ill? This should at least be discussed as another important bias/ limitation.
Response : We acknowledge that longer ICU stays inherently increase the chance of CMV DNAemia detection despite similar frequency of testing (0.47 test/week [0.3;0.7] vs 0.44 test/week [0.29;0.71], p = 0.67). This limitation is discussed as a potential confounder in the limitation of our study.
“Fourth, there is a potential bias introduced by longer ICU stays in CMV-positive patients. Despite an equal frequency of CMV testing between groups, longer ICU stay increase the likelihood of reactivation due to sustained critical illness and immunosuppression. This confounding factor may partly explain the association between CMV DNAemia and longer ICU stays observed in our cohort”
3) If feasible a bit more information (or visualization) on the viral load levels (peak, duration of DNAeamia) over time would be very interesting, to evaluate how significant the reactivations were. Is it possible to stratify the patienst not only according to negative vs. positive DNAemia but also in high and low CMV-DNAemia?
Response : We thank you for this interesting suggestion. As it was also suggested by reviewer 1, we have now added the comparison of three subgroups (absence of CMV DNAemia, CMV DNAemia < 500 IU/mL, and CMV DNAemia > 500 IU/mL) in the Supplementary Material: Table 2. Despite a seemingly well-balanced population across the three groups, there remains a difference in the use of dexamethasone between the group with CMV DNAemia > 500 IU/mL and the group with no CMV DNAemia (28/30 vs. 37/52, p = 0.017), as well as between the group with CMV DNAemia > 500 IU/mL and the group with CMV DNAemia < 500 IU/mL (28/30 vs. 40/53, p = 0.042). Moreover, other main messages of the article is reflected in the subanalysis of these three subgroups: there is no significant difference in mortality between the three subgroups. However, there is an association between the development of CMV DNAemia and the duration of ventilation and hospitalisation, which is particularly marked in the group with CMV DNAemia > 500 IU/mL.
We have added the following statement in the results section:
"The study of three distinct subgroups—absence of CMV DNAemia, CMV DNAemia < 500 IU/mL, and CMV DNAemia > 500 IU/mL—yields findings consistent with those previously reported (Supplementary Material, Table 2)."
- I understood that there are only little data on BAL-fluid PCR results and these were thus not added to statistics?
Response : Thank you for this clarification.
BAL were performed very rarely at the beginning of the pandemic due to limited knowledge about the disease and the perceived additional risk of transmission to healthcare personnel. It was only later, with a better understanding of COVID, that we started performing them, primarily using multiplex PCR without precise quantification of the viral load. Additionally, some samples were also collected from endotracheal aspirations. Given the variability in the types of samples used and the timing during the pandemic when they were collected, the data available on this respiratory samples from our COVID patient cohort are therefore not highly reliable.
Reviewer 3 Report
Comments and Suggestions for Authors
Thank you for the submission, this article is interesting, but there are some comments.
1. There were typing errors in this article.
a. In line 52, the study was approved by the “ethical committee” of Cliniques Universitaires Saint-Luc. The word “ethical committee" should be changes to “Ethics Committee of”
b. In line 73, with a median highest value for CMV viral load of “4366IU/mL”. Give space between 4366 and IU/mL
c. In line 148, question of whether CMV DNAAemia is pathogenic in COVID-19 “disease”. There should be no “disease” word after COVID-19
2. Although in the end it was explained that the clinical outcomes were ICU mortality, ICU length of stay, hospital length of stay, duration of ventilation, duration of ECMO in the results and tables section. It is better if this outcome variables are also explained explicitly in the Methodology Section. In addition, all characteristic variables written in the Results Section should be added to the Methodology Section, such as APACHE II, SOFA, etc.
3. Why did this study use “180 days of ICU admission”, why not 30 days or 90 days?
4. The objective of this study should be the same in the background section and abstract section. Apart from that, the Conclusion Section should also be adjusted to this objective.
5. In my opinion, there are several limitations in this article other than those already written.
a. CMV PCR examination
i. Specimens taken at different times/conditions. In addition, the sample was only carried out once during ICU admission (could cause selection bias)
ii. No serological baseline data. Cannot capture the whole picture of CMV reactivation dynamics (limited interpretation of reactivation onset). This can cause clinical results to be less valid
b. When are APACHE II and SOFA score variables assessed? Does it coincide with the PCR examination? (if there are differences in time and conditions, it can affect the results)
c. The sample size was too small
i. Of 135 patients studied, there were only 30 cases of CMV DNAemia (22%). This may be sufficient for simple univariate analysis, but inadequate for complex multivariate analysis. Additionally, There were many variables included in Cox regression. This can cause a high risk of overfitting and affect the results.
d. Other interventions while in the ICU may have influenced the outcome
6. Suggestions for further research are in accordance with the limitations of the research.

Author Response
Thank you for the submission, this article is interesting, but there are some comments.
- There were typing errors in this article.
- In line 52, the study was approved by the “ethical committee” of Cliniques Universitaires Saint-Luc. The word “ethical committee" should be changes to “Ethics Committee of”
Response : Thank you for this comment, change has been done in the article.
- In line 73, with a median highest value for CMV viral load of “4366IU/mL”. Give space between 4366 and IU/mL
Response : Thank you for this comment, change has been done in the article.
- In line 148, question of whether CMV DNAAemia is pathogenic in COVID-19 “disease”. There should be no “disease” word after COVID-19
Response : Thank you for this comment, change has been done in the article.
- Although in the end it was explained that the clinical outcomes were ICU mortality, ICU length of stay, hospital length of stay, duration of ventilation, duration of ECMO in the results and tables section. It is better if this outcome variables are also explained explicitly in the Methodology Section. In addition, all characteristic variables written in the Results Section should be added to the Methodology Section, such as APACHE II, SOFA, etc
Response : We appreciate this suggestion and have revised the Material and methods section to explicitly describe the clinical outcomes assessed in the study (ICU mortality, ICU and hospital length of stay, etc.). Furthermore, baseline patient characteristics, including APACHE II and SOFA scores, have been detailed in the methodology.
- Why did this study use “180 days of ICU admission”, why not 30 days or 90 days?
Response : Thank you for this interesting question. Shorter time frames such as 30 or 90 days are often used but they might not capture the full spectrum of complications in critically ill patients with severe COVID-19. Indeed, some patients during COVID-19 pandemia remained ventilated and hospitalized in ICU during months.
We selected the 180-day time frame because it allowed for a comprehensive assessment of long-term outcomes in this cohort, including late CMV reactivation and its impact on prolonged ICU stays.
We have clarified this choice in the methodology.
We have added the following statement in the Material and methods section: "The 180-day follow-up period was selected to capture both early and late CMV DNAemia events and assess their impact on long-term outcomes, as shorter follow-up periods may not fully reflect the prolonged disease course of severe COVID-19 patients."
- The objective of this study should be the same in the background sectionand abstract section. Apart from that, the Conclusion Section should also be adjusted to this objective.
Response : We agree and have ensured that the study's objectives are consistent across all sections of the manuscript. Additionally, the conclusion section has been adjusted to directly reflect these objectives.
- In my opinion, there are several limitations in this article other than those already written.
- CMV PCR examination
- Specimens taken at different times/conditions. In addition, the sample was only carried out once during ICU admission (could cause selection bias)
Response : We fully agree that the fact that the samples were not collected in a protocolized manner represents a limitation of this retrospective study. This is acknowledged as part of the limitations detailed in the final section of the discussion. However, the samples were not collected only once during the ICU stay. Indeed, patients had a median of 2 [1;4] tests performed (described in the Results).
- No serological baseline data. Cannot capture the whole picture of CMV reactivation dynamics (limited interpretation of reactivation onset). This can cause clinical results to be less valid
Response : We completely agree with your comment. This limitation is also thoroughly described in the final section of the discussion and has been added to the Materials and Methods section, as suggested by Reviewer 1. For this reason, we use the term CMV DNAemia rather than reactivation in the article, as the pre-admission status is unknown in our cohort, although an acute infection occurring during hospitalization is unlikely in our ICU setting
- When are APACHE II and SOFA score variables assessed? Does it coincide with the PCR examination? (if there are differences in time and conditions, it can affect the results)
Response : These scores were performed at ICU admission for all our patient from our cohort. This information has been added in Material and methods section : “Baseline characteristics such as Acute Physiology and Chronic Health Evaluation II (APACHE II) and Sequential Organ Failure Assessment (SOFA) scores were collected at ICU admission to evaluate patient severity.”
- The sample size was too small
- Of 135 patients studied, there were only 30 cases of CMV DNAemia (22%). This may be sufficient for simple univariate analysis, but inadequate for complex multivariate analysis. Additionally, There were many variables included in Cox regression. This can cause a high risk of overfitting and affect the results.
Response : Thank you for your comment. This has been clarified in the limitation part of the discussion.
- Other interventions while in the ICU may have influenced the outcome
Response : We agree that certain treatments may have influenced the outcomes of our patients. That said, patient-related characteristics and treatment-related factors specific to their condition are relatively uniform between the two groups. The severity of the two populations, as assessed by APACHE II and SOFA scores, also appears to be similar between the two groups. Finally, the monocentric nature of the study ensures that patient management was largely uniform across the cohort.
Suggestions for further research are in accordance with the limitations of the research.
Response : We appreciate this suggestion and have revised the limitations to include specific recommendations for future studies for each limitation. These include longitudinal monitoring of CMV reactivation, baseline serological testing, and larger, multicenter cohorts to validate the finding.
Reviewer 4 Report
Comments and Suggestions for Authors
Manuscript is very clearly written and succinct. Presentation of figures and tables also clear for readers to quickly understand.
Some clarification should be provided in methods regarding the criteria for CMV testing within the ICU unit - i.e. why were the 135 patients tested for CMV out of the whole 188 COVID positive cohort? The information is provided in the limitations section (line 157-167) but should be provided upfront in the methods section also.
A major weakness of the methodology that can be improved with presumably available data is that no comparative %s are given for DNAemia for the rest of the ICU unit (i.e. regardless of COVID infection). Such data for the rest of the ICU unit and their associations with prolonged duration of IMV, ECMO or LOS would be important to help place the currently reported findings within the COVID positive patient group in context. Are these reported associations of CMV DNAemia with increased ICU and hospital LOS, duration of IMV +/- ECMO agnostic to COVID status?
The writing is careful not to make any unsupported conclusions from the currently provided data. The authors' main conclusion that CMV DNAemia is "common in patients under IMV for severe SARS-COV2 pneumonia" may be supported by the data provided (i.e. that 22% of their SARS CoV2 patients undergoing IMV were CMV DNAemia >500IU/mL). However it is hard to interpret the relevance of this finding, and the manuscript should be improved by providing more context and expanding the analysis to non-COVID patients.
Author Response
Some clarification should be provided in methods regarding the criteria for CMV testing within the ICU unit - i.e. why were the 135 patients tested for CMV out of the whole 188 COVID positive cohort? The information is provided in the limitations section (line 157-167) but should be provided upfront in the methods section also.
Response : Thank you for your comment. As suggested by you and the other reviewers, we have added these details to the Materials and Methods section.
A major weakness of the methodology that can be improved with presumably available data is that no comparative %s are given for DNAemia for the rest of the ICU unit (i.e. regardless of COVID infection). Such data for the rest of the ICU unit and their associations with prolonged duration of IMV, ECMO or LOS would be important to help place the currently reported findings within the COVID positive patient group in context. Are these reported associations of CMV DNAemia with increased ICU and hospital LOS, duration of IMV +/- ECMO agnostic to COVID status?
Response : We acknowledge this important point. Unfortunately, this study specifically focused on the severe SARS-CoV-2 cohort, and data for non-COVID ICU patients were not collected for this analysis. However, we agree that comparative data from non-COVID patients could provide meaningful context and would improve interpretation of the findings. This subject has been discussed in the Discussion par. For example in a previous study involving patients under MV for ARDS unrelated to COVID, CMV reactivation was found in 18,4% (74/401) of patients. Classically in the literature, CMV reactivation in the non-COVID-19 population has been associated with longer duration of ventilation and longer ICU stay.
The writing is careful not to make any unsupported conclusions from the currently provided data. The authors' main conclusion that CMV DNAemia is "common in patients under IMV for severe SARS-COV2 pneumonia" may be supported by the data provided (i.e. that 22% of their SARS CoV2 patients undergoing IMV were CMV DNAemia >500IU/mL). However it is hard to interpret the relevance of this finding, and the manuscript should be improved by providing more context and expanding the analysis to non-COVID patients.
Response : We have modified the conclusion to emphasize the association between CMV DNAemia and prolonged ICU outcomes within the studied SARS-CoV-2 cohort without overgeneralizing the results. Additionally, we have highlighted the need for further studies comparing CMV reactivation rates and outcomes in COVID-19 and non-COVID ICU patients to better understand the relevance of this finding.